

# Effects of particle size of ground alfalfa hay on caecal bacteria and archaea populations of rabbits

Mei Yuan, Siqiang Liu, Zhisheng Wang, Lizhi Wang, Bai Xue, Huawei Zou, Gang Tian, Jingyi Cai and Quanhui Peng

Animal Nutrition Institute, Key Laboratory of Bovine Low-Carbon Farming and Safety Production, Sichuan Agricultural University, Chengdu, China

## ABSTRACT

This work was aimed to investigate the effects of the different particle size of ground alfalfa hay on caecal microbial and archeal communities of rabbits. One hundred-twenty New Zealand rabbits ($950.3 \pm 8.82$ g) were allocated into four treatments, with five replicates in each treatment and six rabbits in each replicate. The particle sizes of the alfalfa meal in the four treatment diets were 2,500, 1,000, 100 and 10 $\mu$m respectively, while the other ingredients were ground through a 2.5 mm sieve. High-throughput sequencing technology was applied to examine the differences in bacteria and methanogenic archaea diversity in the caecum of the four treatment groups of rabbits. A total of 745,946 bacterial sequences (a mean of $31,081 \pm 13,901$ sequences per sample) and 539,227 archaeal sequences (a mean of $22,468 \pm 2,443$ sequences per sample) were recovered from twenty-four caecal samples, and were clustered into 9,953 and 2,246 OTUs respectively. A total of 26 bacterial phyla with 465 genera and three archaeal phyla with 10 genera were identified after taxonomic summarization. Bioinformatic analyses illustrated that Firmicutes ($58.69\% \sim 68.50\%$) and Bacteroidetes ($23.96\% \sim 36.05\%$) were the two most predominant bacterial phyla and Euryarchaeota (over 99.9%) was the most predominant archaeal phyla in the caecum of all rabbits. At genus level, as the particle size of alfalfa decreased from 2,500 to 10 $\mu$m, the relative abundances of *Ruminococcaceae UCG-014* ($P < 0.001$) and *Lactobacillus* ($P = 0.043$) were increased and *Ruminococcaceae UCG-005* ($P = 0.012$) was increased first and then decreased when the alfalfa particle size decreased, while *Lachnospiraceae NK4A136 group* ($P = 0.016$), *Ruminococcaceae NK4A214* ($P = 0.044$), *Christensenellaceae R-7 group* ($P = 0.019$), *Lachnospiraceae other (Family)* ($P = 0.011$) and *Ruminococcaceae UCG-013* ($P = 0.021$) were decreased. The relative abundance of *Methanobrevibacter* was increased from 62.48% to 90.40% ($P < 0.001$), whereas the relative abundance of *Methanosphaera* was reduced from 35.47% to 8.62% ($P < 0.001$). In conclusion, as the particle size of alfalfa meal decreased, both the bacterial and archaeal population in the caecum of rabbit experienced alterations, however archaea response earlier than bacteria to the decrease of alfalfa meal particle size.

Corresponding author
Quanhui Peng,
pengquanhui@126.com

## INTRODUCTION

Rabbit meat is an important part of meat products in China. According to FAO official database FAOSTA (http://faostat.fao.org), the output of Chinese rabbit meat was $7.27 \times 10^5$ $t$ and the per capita consumption was 0.527 kg in 2013. In recent years, the rabbit breeding industry has also prospered, and more and more attention has been paid to rabbit research. Rabbit is a monogastric herbivore animal that has a certain ability to digest fiber. Although the digestibility is not as high as other herbivores (*Voris et al., 1940*; *Slade & Hintz, 1969*), plant fiber has special nutritional and physiological functions in rabbits (*Chiou, Yu & Lin, 1994*; *Cheeke, 1987*; *Jenkins, 1999*). Undoubtedly, dietary fiber is the most important component in the diet of rabbits, especially, alfalfa as a balanced source of fiber is a good choice for rabbit feed (*García et al., 1995a*; *García et al., 1995b*). Rabbits possess very developed caecum, accounting for about 40% capacity of the gastrointestinal tract (*Jenkins, 1999*). The fiber in the diet of rabbit is mainly degraded by caecal microorganisms such as bacteria, fungi, archaea, etc. The species, quantity and balance of intestinal microorganisms are important indicators of the health of animals, and are also important manifestations of function of digestive tract (*Nicholson et al., 2012*; *Jami et al., 2013*). Thus, it is very important to maintain the stable structure of the intestinal microorganisms in rabbits for the digestion and absorption of nutrients and intestinal health.

The intestinal microorganism is always in the dynamic change, and there are many factors affect the composition of the gut microbiota, especially fiber plays an important role in the balance of intestinal microflora structure. Previous work (*García et al., 2000*; *Chang et al., 2006*) showed that different fiber sources had effects on the caecel microorganism fermentation activity in rabbits. *Cao et al. (2016)* also suggested that fiber source will change the methanogenic community in the hindgut of pigs. Whereas the fiber level has controversial effects on the changes of caecal microorganism in rabbits. *Boulahrouf, Fonty & Gouet (1991)* showed that when dietary fiber level increased, the number of bacteria decomposing fibers in caecum of rabbits increased. On the contrary, *Bennegadi et al. (2003)* showed that decreased dietary fiber levels reduced the proportion of archaea community but increased the number of *Bacteroidetes* and *Ruminococcus albus* (both bacteria can break down the fiber) in the caecum of rabbits. Physical structure, especially particle size, is another important characteristic of fiber that influences caecal microorganism in rabbits. *García et al. (2000)* reported that the feed particle size has a significant effect on caecum microorganism fermentation in rabbits, however, specific studies on changes in microbial composition and structure are scarce.

In addition, most of the existing reports about particle size of diet fiber in rabbits are descriptions on the retention time of chyme in the caecum, the digestion of nutrients and the growth performance (*Gidenne et al., 1991*; *Gidenne, 1993*; *Nicodemus et al., 2010*; *Romero et al., 2011*). Yet limited studies on the effects of fiber particle size on caecal microbial composition in rabbits exist, let alone comparison of the sensitivity of bacteria and archaea to the diet fiber particle size. Therefore, the objective of this study was to explore the effects of different alfalfa hay particle size on the composition and response of caecum bacteria and archaea in rabbits using Illumina sequencing technology.

## MATERIALS AND METHODS

The experimental protocol used in the present study was approved by the Animal Policy and Welfare Committee of the Agricultural Research Organization of Sichuan Province, China, and was in accordance with the guidelines of the Animal Care and Ethical Committee of the Sichuan Agricultural University (Permission code SYXK (chuan) 2014-187).

### Animal experiment and sample collection

A total of 120 New Zealand rabbits (half male and female) with average body weight (950.3 ± 8.82 g), weaned at 35 days of age were selected and allocated into 4 treatments, with five replicates in each treatment and six rabbits in each replicate, and the rabbits were kept in three pairs with two rabbits per cage. Rabbits were housed in the same building (Sichuan Agricultural University, China) in flat-deck cages (600 × 250 × 330 mm) for the 49 d experiment. The temperature inside the house was maintained at 15–25 °C and rabbits had *ad libitum* access to water and diets during the whole experimental period. Neither feed nor drinking water was medicated with antibiotics, but a coccidiostatic (robenidine) was provided in the feed. In this study, the diets (Table S1) were formulated according to NRC (1977) Growing Rabbit Feeding Standards, and the alfalfa meal incorporated into the four diets were 2,500, 1,000, 100 and 10 μm, respectively. Firstly, alfalfa meal with particle sizes of 2,500, 1,000, 100 and 10 μm were produced, and then the rest of the ingredients were milled through a 2.5-mm grinder screen. After all the ingredients were ready, they were mixed and granulated (diameter was 3 mm). After the finish of the growth experiment, 24 animals were slaughtered (six rabbits per diet) by cervical dislocation 1 h before dark (19:00) to avoid soft feces excretion. Once slaughtered, 50 g of cecal content was collected in sterile conditions. The samples were immediately frozen at −80 °C until DNA extraction.

### DNA extraction

Caecal samples were thawed at room temperature and kept on ice during the extraction process. According to the manufacturer's instructions, total genomic DNA was extracted from caecal samples using a DNeasy PowerSoil Kit (Qiagen, Valencia, CA, USA) that included a step of the microbiological cells to be mechanically broken. The quality and concentration of the extracted DNA was detected, respectively, using 0.8% agarose gel electrophoresis and a NanoDrop ND-1,000 spectrophotometer (Nyxor Pharmacia, Paris, France). All extracted DNA samples were diluted to 10 ng/μL using sterile ultrapure water and stored at −20 °C until used for real-time PCR and Illumina sequence analyses.

### PCR amplification, Illumina library generation and sequencing

The V4 variable region of 16S rRNA gene was amplified by using caecal total DNA as template. Selecting 515F (5′-GTGCCACMCCGCGGTAA-3′) and 806R (5′-GGACTACHVGGGTWTCTAAT-3′) as the primer pair for the bacteria (*Caporaso et al., 2011*), while selecting A516F (5′-TGYCAGCCGCCGCGGTAAHACCVGC-3′) and U806R (5′-GGACTACHVGGGTWTCTAAT-3′) as the primer pair for the archaea (*Kuroda et al., 2015*). 16S rRNA genes were amplified using the specific primer with 12 nt unique barcode, and used the same reaction system for PCR amplification for bacteria and archaea. The total

PCR mixture (25 μL) contained 1 × PCR buffer, 1.5 mM MgCl$_2$, each deoxynucleoside triphosphate at 0.4 μM, each primer at 1.0 μM, 0.5 U of KOD-Plus-Neo (Toyobo, Tokyo, Japan) and 10 ng template DNA. The PCR amplification program consists of initial denaturation at 94 °C for 1 min, followed by 30 cycles of denaturation at 94 °C for 20 s, annealing at 54 °C for 30 s, elongation at 72 °C for 30 s, and a final extension at 72 °C for 5 min. Three replicates of PCR reactions for each sample were combined together, PCR products mixed with 1/6 volume of 6× loading buffer were loaded on 2% agarose gel for detection. We used a 2 step process for library preparation. First, 16S amplicons were generated using PCR to generate 200–400 bp amplicons. Second, we added unique barcodes to samples using emulsion PCR (EmPCR) to prevent chimera formation (*Williams et al., 2006*), and the PCR was stopped at linear stage. This methodological basic principle is dilution and compartmentalization of template molecules in water droplets in a water-in-oil emulsion. Ideally, the dilution is to a degree where each droplet contains a single template molecule and functions as a micro-PCR reactor (*Kanagal-Shamanna, 2016*). This method reduces the formation of artifactual molecules, as often seen in conventional PCR, thus preserving library complexity (*Williams et al., 2006*; *Qiu et al., 2001*). The electrophoresis band was purified using OMEGA Gel Extraction Kit (Omega Bio-Tek, USA), the gel purified barcoded amplicons were pooled with equal molar amount and quantified on a Qubit@ 2.0 Fluorometer (Thermo Scientific). Finally, 15% of PhiX control library was spiked into the amplicon pool to improve the unbalanced and biased base composition. In brief, sequencing libraries were generated using TruSeq DNA PCR-Free Sample Prep Kit following manufacturer's recommendations and index codes were added, and using ZymoBIOMICS Microbial Community Standard (Cat#D6300) as the positive control. The library quality was assessed on the Qubit@ 2.0 Fluorometer (Thermo Scientific) and Agilent Bioanalyzer 2100 system. At last, the library was applied to paired-end sequencing (2 × 250 bp) with the Hiseq Illumina Sequencing Platform (Rhonin Biosciences Co., Ltd., Chengdu, China), according to the protocols described by *Caporaso et al. (2012)*.

## Bioinformatics analysis

The sequences were analyzed according to Usearch (version 7.1 http://drive5.com/uparse/) and QIIME (*Caporaso et al., 2010a*) pipeline. Paired-end reads from the original DNA fragments were merged using FLASH (*Magoč & Salzberg, 2011*). Then sequences were assigned to each sample according to the unique barcode. The first step was to filter low quality reads (length < 200 bp, more than two ambiguous base 'N', or average base quality score < 30) and truncated sequences where quality scores decay (score < 11) using Trimmomatic (*Bolger, Lohse & Usadel, 2014*) and Usearch. After finding duplicated sequences, discarded all the singletons, which may be sequencing errors (http://www.drive5.com/usearch/manual/singletons.html) and lead to false positive results for overestimation of diversity. Sequences were clustered into operational taxonomic units (OTUs) at 97% identity threshold using UPARSE algorithms (*Edgar, 2013*), and picked representative sequences and removed potential chimeras using Uchime algorithm (*Edgar et al., 2011*). Taxonomy were assigned using the Silva (https://www.arb-silva.de) database (*Quast et al., 2013*) and uclust classifier in QIIME (version 1.8.0). Representative
sequences were aligned using PyNAST (*Caporaso et al., 2010b*) embedded in QIIME. After quality checking, phylogenetic trees were reconstructed based on maximum likelihood–approximation method using the generalised time-reversible (GTR) model in FastTree (*Price, Dehal & Arkin, 2010*).

Then the filtered OTU is removed from the OTU table in the process of evolutionary tree reconstruction, and the OTU table is resampled so that each sample has the same sequence number. Phylogenetic diversity (Faith's PD18) was calculated using Picante (*Kembel et al., 2010*). Weighted and Unweighted Unifrac distances were calculated in GUniFrac version 1.1 (*Chen et al., 2012*). The analysis of alpha and beta diversity metrics was conducted with Vegan (version 2.0-2. R CRAN package). Rarefaction curves were generated based on these three metrics. Principal component analysis (PCA) was applied to reduce the dimensions of original community data. Hierarchical cluster analysis was done using R (https://www.R-project.org/) function hclust. To identify if there were significant differences among different groups, permutational multivariate analysis of variance was performed based on the dissimilarity matrix.

### Statistical analysis

The effects of alfalfa meal particle size on cecal microbial communities were analyzed using the MIXED procedure of SAS v9.4 (SAS Institute Inc., Cary, NC, USA). The model used for the analysis was

$$Y_{ij} = \mu + T_i + e_{ij}$$

where $Y_{ij}$ is an observation on the dependent variable ij, $\mu$ is the overall population mean, $T_i$ isthe fixed effect of alfalfa meal particle size, and $e_{ij}$ is the random error associated with the observation ij. Tukey-Kramer multiple comparison tests were performed after differences were detected. Differences between means with $P < 0.05$ were accepted as statistically significant differences.

## RESULTS

### Sequencing depth, coverage and alpha diversity

A rarefaction test was performed at the OTU level and the results are presented (Fig. S1). As can be seen from the rarefaction curve that all of the samples tended to reach a plateau, indicating that the sequencing quantity of both bacteria and archaea covered most microorganisms. Measures of alpha diversity were shown in Table 1. As for the bacteria, the OTUs, Chao1 index, Shannon-Wiener index and PD value (phylogenetic diversity) were all numerically higher in group 10 $\mu$m than that of the other three groups ($P > 0.05$). As for the archaea, the Shannon-Wiener index of group 1,000, 100 and 10 $\mu$m were significantly decreased compared with group 2,500 $\mu$m ($P = 0.044$). However, no significant difference was obtained among 1,000, 100 and 10 $\mu$m groups. The above results suggested that rabbit caecum alpha diversity of bacteria and archaea experienced different alterations when the alfalfa particle size decreased.

A total of 745,946 bacterial sequences were generated after quality control with a mean of $31,081 \pm 13,901$ (mean $\pm$ standard deviation [SD], $n = 24$) per sample, and the mean

**Table 1** The average value of alpha diversity index of caecum microbes of rabbits fed diets with different particle size of alfalfa meal.

| Item[1] | Treatments[2] | | | | SEM | P-value[3] |
|---|---|---|---|---|---|---|
| | 2,500 μm | 1,000 μm | 100 μm | 10 μm | | |
| Bacteria | | | | | | |
| OTUs | 1961.33 | 1784.33 | 1815.67 | 2022.83 | 51.313 | 0.305 |
| Chao1 value | 3331.69 | 2952.54 | 2878.53 | 3373.24 | 142.157 | 0.519 |
| Shannon value | 6.07 | 6.03 | 6.12 | 6.35 | 0.054 | 0.160 |
| PD value | 164.72 | 151.87 | 154.46 | 168.19 | 3.469 | 0.228 |
| Archaea | | | | | | |
| OTUs | 363.17 | 340.17 | 363.50 | 357.00 | 4.382 | 0.198 |
| Chao1 value | 728.05 | 716.41 | 721.69 | 728.00 | 8.213 | 0.957 |
| Shannon value | 2.27[a] | 2.16[b] | 2.14[b] | 2.17[b] | 0.349 | 0.044 |
| PD value | 18.42 | 17.63 | 18.40 | 18.31 | 5.809 | 0.681 |

**Notes.**
[1] Data are the means of six replicates.
[2] The particle size of alfalfa meal was 2,500, 1,000, 100 and 10 μm, respectively.
[3] Values with different superscripts in the same row mean significant difference ($P < 0.05$).

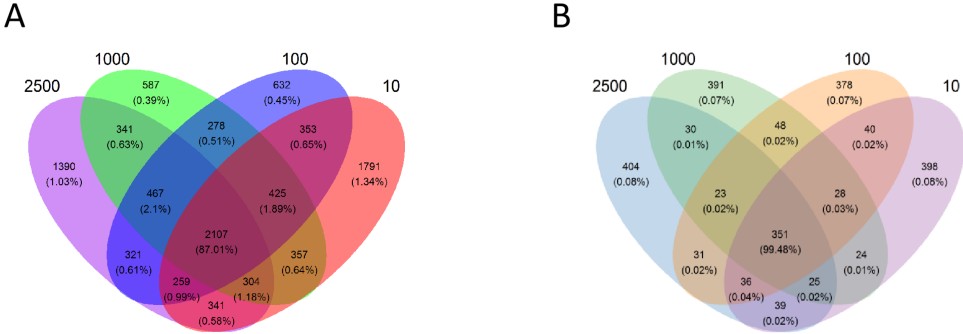

**Figure 1** Venn diagram representation of the shared and exclusive bacterial (A) and archaeal (B) OTUs at 97% similarity level of the four treatment groups. The percentage data in parentheses is the sequence abundance of the corresponding OTUs out of the total OTU.

length of tag N50 is 304 ± 6 base pairs (bp). Based on the principle that the similarity is greater than 97%, the obtained effective sequences were clustered into a total of 9,953 OTUs, the average OTU was 1,896 ± 251 (mean ± SD, $n = 24$) per sample. A total of 2107 OTUs were shared in all of four treatments, while 1,390, 587, 632 and 1,791 OTUs were exclusively found in group 2,500, 1,000, 100 and 10 μm, respectively (Fig. 1A). Illumina HiSeq sequence of the archaeal 16S rRNA yielded 539,227 sequences for the 24 caecal samples, with a mean of 22,468 ± 2,443 sequences per sample, and the mean length of tag N50 is 292 ± 4. A total of 2246 OTUs were assigned, and each sample contained 336 ± 21 OTUs. Furthermore, 351 OTUs were shared by the four groups, while 404, 391, 378 and 398 OTUs were exclusively found in group 2,500, 1,000, 100 and 10 μm, respectively (Fig. 1B).

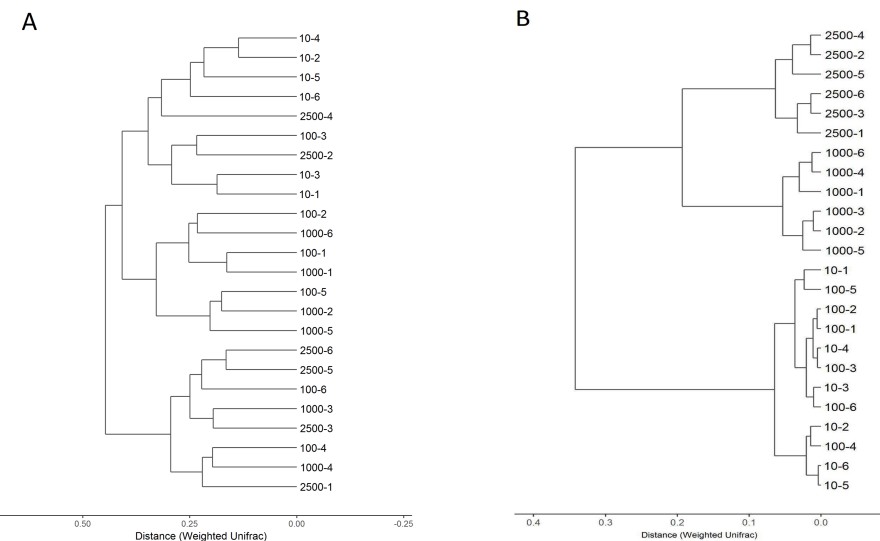

**Figure 2** **Hierarchical clustering of bacterial (A) and archaeal (B) communities assessed using weighted UniFrac metric analysis of OTUs at 97% similarity.** The scale bar shows approximate weighted UniFrac metric similarity coefficient of 0.25 in bacteria, the archae group 2,500 μm = tag number 2,500-1, 2,500-2, 2,500-3, 2,500-4, 2,500-4, 2,500-5 and 2,500-6; group 1,000 μm = tag number 1,000-1, 1,000-2, 1,000-3, 1,000-4, 1,000-5 and 1,000-6; group 100 μm = tag number 100-1, 100-2, 100-3, 100-4, 100-5 and 100-6; group 10 μm = tag number 10-1, 10-2, 10-3, 10-4, 10-5 and 10-6.

## Analysis of microbial community similarity in caecum of rabbits

Community OTU comparisons were visualised by clustering analysis (OTU ≥ 97% identity, species level similarity) using weighted unifrac clustering in Fig. 2. The closer samples and the shorter branches are indicating the more similar the species composition of the two samples. The relationship of bacterial community between different treatments showed that the samples were clustered into three branches (Fig. 2A). All samples in 10 μm group clustered into the same clade, however, sample 2,500-2, 2,500-4 and 100-3 were clustered into this clade. The samples of group 2,500, 1,000 and 100 μm were blended together. This means that the bacterial composition changed when the particle size of alfalfa meal decreased from 100 to 10 μm. Similarly, a difference was also observed in the composition of archaea between the four treatments (Fig. 2B). The 2,500 and 1,000 μm groups were clustered thoroughly into one clade first and then mingled together, and the distance was larger than 0.2 between the two groups. The 100 and 10 μm groups mixed together and could not be distinguished, and the distance was less than 0.1. This means archaea is kept stable when the particle size of alfalfa meal larger than 1,000 μm, and it is altered when the particle size meal is lowered less than 100 μm. In addition, the archaea in the caecum of rabbit is changed earlier than bacteria as alfalfa meal particle size decreased.

A PCoA plot of overall diversity based on weighted UniFrac metric was generated (Fig. 3). The closer the distance between points means the more similar the community structure of the two samples. The PCoA plot demonstrated that the bacterial community between 10 μm group and other three treatments could be clearly distinguished ($P < 0.01$). However, the bacterial community among 2,500, 1,000 and 100 μm groups could not be

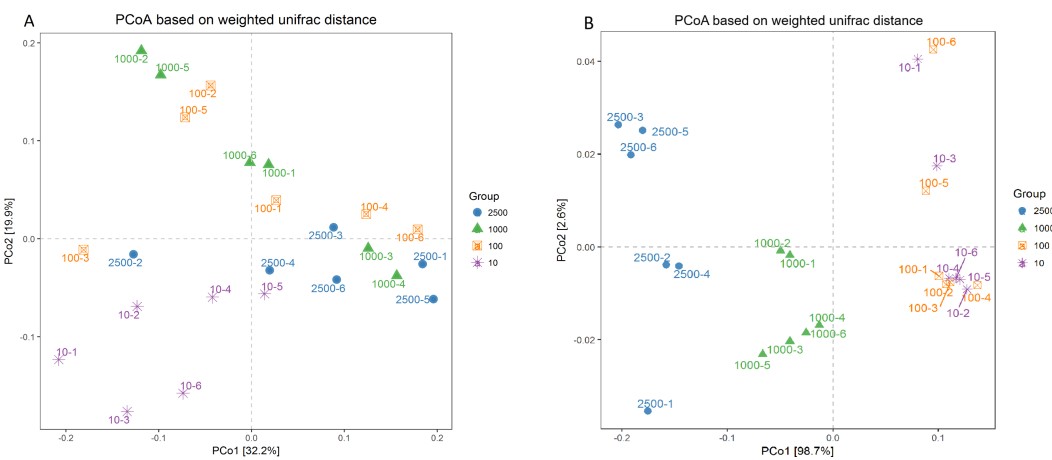

**Figure 3** **Principal co-ordinate analysis (PCoA) scores plot generated from rabbits caecum sample by a weighted UniFrac analysis at the 97% similarity level.** Bacteria (A) and archaea (B).

obvious isolated ($P > 0.05$), and PCo1 accounted for 32.2% (Fig. 3A; Table S2). In addition, the six archaeal samples of group 100 and 10 μm were mixed together ($P = 0.995$) but could be separated thoroughly from group 2,500 and 1,000 μm ($P = 0.025$), and the PCo1 accounted for 98.7% (Fig. 3B; Table S2). And the same result was obtained in our unweighted UniFrac metric analysis (Fig. S3). This again, proved that the archaeal composition structure changed before the bacterial structure, and the uniformity of archaea is better with the decrease of alfalfa meal particle size.

## Taxonomy of rabbit caecum microbial composition

The sequences in the present experiment were finally annotated as bacteria and archaea, a total of 26 bacterial phyla and 3 archaeal phyla are identified. As for the bacteria, among which 23 phyla were detected in the 2,500 and 100 μm groups, and 21 and 24 phyla were obtained in 1,000 and 10 μm group, respectively. Figure 4 presents the relative abundance of microbial composition at the phylum level of different treatments. There were only 3 bacterial phyla with relative abundance larger than 1% in 2,500, 1,000 and 100 μm groups (Fig. 4A), and the two most abundant phyla were Firmicutes (58.69 ± 0.114%, 67.77 ± 0.105% and 67.82 ± 0.097%) and Bacteroidetes (36.05 ± 0.118%, 26.24 ± 0.114% and 26.20 ± 0.105%), followed by Proteobacteria (2.69 ± 0.003%, 3.37 ± 0.005% and 3.40 ± 0.004%). At the meanwhile, the 10 μm group had 4 bacterial phyla with relative abundance above 1%. In addition to the three dominating phyla, Firmicutes (68.50 ± 0.052%), Bacteroidete (23.96 ± 0.056%) and Proteobacteria (3.34 ± 0.002%), the relative abundance of Tenericutes (2.08 ± 0.012%) was also greater than 1%. The archaeal abundance analysis showed that the highest relative abundance was Euryarchaeota (over 99.9%) in all of the four treatment groups (Fig. S2). The remaining archaeal abundance (Thaumarchaeota and Miscellaneous Crenarchaeotic Group) were less than 0.1%.

At the genus level, the bacteria kingdom is composed of 465 genera, whereas only 26 genera with relative abundance larger than 1%, among which 10 of them were unclassified.

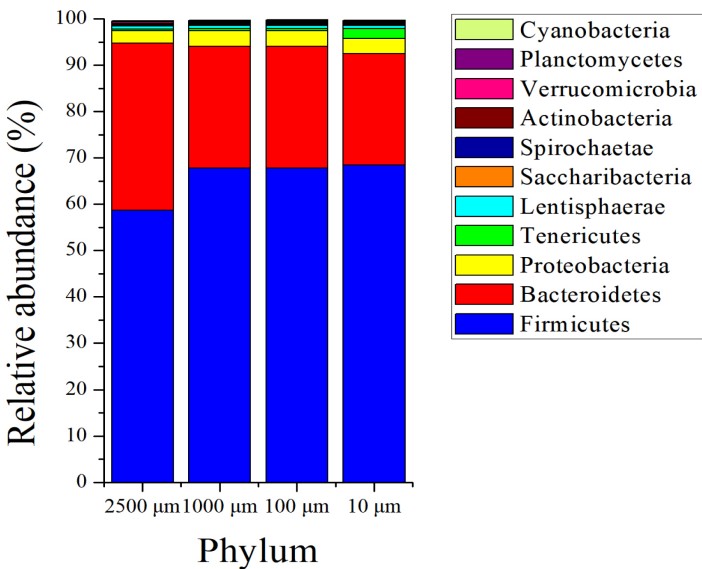

**Figure 4** **Phylum level composition of bacteria.** A color-coded bar plot shows the average relative abundance of bacterial phyla (>0.1%) distribution in different treatment groups.

The number of genera with relative abundance greater than 1% in the four treatment groups were 19 (2,500 μm), 23 (1,000 μm), 22 (100 μm) and 20 (10 μm), respectively. The most abundant genera (relative abundance more than 4%) were Unclassified *Bacteroidales S24-7 group* (20.39 ± 0.112%), *Ruminococcaceae UCG-014* (13.25 ± 0.101%), *Lachnospiraceae NK4A136 group* (10.73 ± 0.065%) and *Rikenellaceae RC9 gut group* (5.00 ± 0.066%) in the 2,500 μm group. Unclassified *Bacteroidales S24-7 group* (14.48 ± 0.087% and 14.56 ± 0.088%), *Ruminococcaceae UCG-014* (9.34 ± 0.040% and 11.55 ± 0.097%), *Lachnospiraceae NK4A136 group* (8.80 ± 0.046% and 9.61 ± 0.035%), unclassified *Clostridiales vadinBB60 group* (5.72 ± 0.060% and 4.86 ± 0.015%) and *Ruminococcaceae NK4A214 group* (4.71 ± 0.024% and 4.39 ± 0.027%) were the most two abundant genera in 1,000 and 100 μm groups. In the 10 μm group, *Ruminococcaceae UCG-014* (33.82 ± 9.752%) and Unclassified *Bacteroidales S24-7* (11.81 ± 2.876%) group were the two most abundant genera. The relative abundance of the 10 unclassified genera accounted for 29.45%, 20.55%, 30.34% and 24.86% in 2,500, 1,000, 100 and 10 μm groups, respectively (Fig. 5A).

Table 2 shows the significantly affected bacteria by the decrease of alfalfa particle size. *Ruminococcaceae UCG-014* ($P < 0.001$) and *Lactobacillus* ($P = 0.043$) were increased while the alfalfa particle size decreased, while the relative abundance of *Lachnospiraceae NK4A136 group* ($P = 0.016$), *Ruminococcaceae NK4A214* ($P = 0.044$), *Ruminococcaceae UCG-005* ($P = 0.012$), *Christensenellaceae R-7 group* ($P = 0.019$), *Lachnospiraceae other* (*Family*) ($P = 0.011$) and *Ruminococcaceae UCG-013* ($P = 0.021$) were decreased.

Meanwhile, a total of 10 genera were assigned from the sequences of archaea, however only 4 genera were classified. Archaeal classification at the genus level demonstrated that the dominating genus in caecum of rabbits was *Methanobrevibacter*, and its relative abundance was significantly increased from 62.48% to 90.40% as the particle size of alfalfa decreased

A

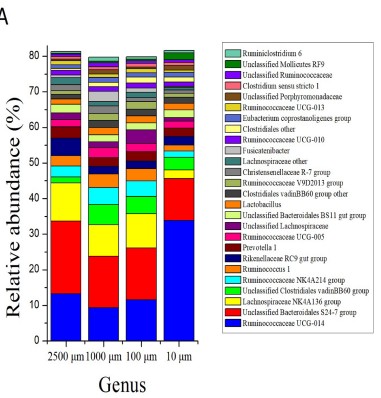

B

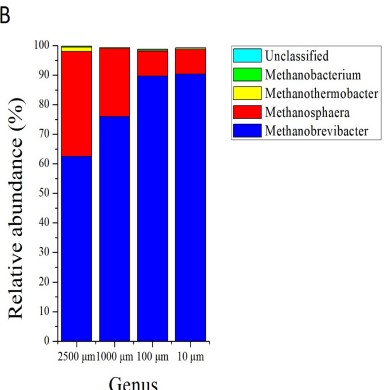

**Figure 5 Genus level composition.** Bar plots show average relative abundance (%) of bacterial (A) and archaeal (B) in different particle sizes. Only bacterial genera with relative abundance more than 1% are shown.

from 2,500 to 10 µm ($P < 0.001$). The *Methanosphaera* was the second largest genus in the caecum of rabbits, and its relative abundance decreased from 35.47% to 8.26% with the decrease of alfalfa particle size ($P < 0.001$) (Fig. 5B; Table 2).

# DISCUSSION

In the present work, we attempted to study the impact of different fiber particle size on the caecal microflora of rabbits. Variations in the microbial community structure of caecum in rabbits were observed according to the decrease of particle size of alfalfa hay, and the archaea responsed earlier than bacteria when the particle size of alfalfa decreased.

Effects of fiber particle size on growth performance and digestibility of nutrients in rabbit has been assessed in previous studies (*Gidenne et al., 1991*; *Romero et al., 2011*; *Liu et al., 2018*); however, to the best of our knowledge, this is the first implementation of high throughput sequencing technology to investigate the relationship between fiber particle size and caecal microflora of rabbits. We found that the alpha diversity index of the finest particle size was the highest numerically, although there was no significant difference in bacterial diversity in this study. We have previously confirmed that the growth performance of rabbits at group 10 µm particle size is the best (*Liu et al., 2018*). There are also studies showing that high diversity of microorganisms is beneficial to animals (*Arrazuria et al., 2016*). However, we observed the Shannon index of archaea decreased significantly with the decrease of particle size, which was contrary to the change of ADG (average daily gain) and ADFI (average daily feed intake) in rabbits. This may be due to the low content of archaea (0.2 ± 0.23%) in the cecum (*Liu et al., 2016*) and its change in diversity was insufficient to affect the growth performance of rabbits.

We also discovered that microbial abundance varied with the different particle size of alfalfa, this is consistent with previous studies on poultry (*Engberg et al., 2004*; *Santos et al., 2006*). Furthermore, bacteria belonging to the phylum Firmicutes dominated the bacterial community of rabbit caecum in all of the four treatments (58.69% ~68.50%), and these

**Table 2  Phyla and genera with different relative abundance in caecum of rabbits fed diets with different particle size of alfalfa meal.**

| Item[1] | Treatments[2] | | | | SEM | *P*-value[3] |
|---|---|---|---|---|---|---|
| | 2,500 μm | 1,000 μm | 100 μm | 10 μm | | |
| **Bacteria phylum** | | | | | | |
| Proteobacteria | 2.69[b] | 3.37[a] | 3.40[a] | 3.34[a] | 0.09 | 0.005 |
| Tenericutes | 0.35[b] | 0.48[b] | 0.47[b] | 2.11[a] | 0.193 | <0.001 |
| Cyanobacteria | 0.04[b] | 0.18[a] | 0.12[ab] | 0.16[a] | 0.019 | 0.042 |
| Fusobacteria | 0.03[b] | 0.02[b] | 0.04[ab] | 0.07[a] | 0.006 | 0.022 |
| SHA-109 | 0[b] | 0[b] | 0[b] | 0.005[a] | 0.000 | 0.010 |
| **Bacterial genus** | | | | | | |
| *Ruminococcaceae UCG-014* | 13.25[b] | 9.34[b] | 11.55[b] | 33.82[a] | 2.642 | <0.001 |
| *Lachnospiraceae NK4A136 group* | 10.73[a] | 8.80[a] | 9.61[a] | 2.40[b] | 1.071 | 0.016 |
| *Ruminococcaceae NK4A214 group* | 3.13[ab] | 4.71[a] | 4.39[a] | 1.74[b] | 0.428 | 0.044 |
| *Ruminococcaceae UCG-005* | 2.10[b] | 2.84[a] | 2.38[ab] | 1.99[b] | 0.106 | 0.012 |
| *Lactobacillus* | 1.55[b] | 1.99[a] | 1.90[a] | 1.87[a] | 0.06 | 0.043 |
| *Christensenellaceae R-7 group* | 1.68[ab] | 2.14[a] | 1.12[b] | 1.01[b] | 0.153 | 0.019 |
| *Lachnospiraceae other* (*Family*) | 1.89[a] | 1.27[ab] | 1.54[a] | 0.72[b] | 0.137 | 0.011 |
| *Ruminococcaceae UCG-013* | 1.15[a] | 1.00[a] | 1.12[a] | 0.67[b] | 0.064 | 0.021 |
| *Clostridium sensu stricto 1* | 0.69[b] | 0.91[a] | 1.00[a] | 0.83[ab] | 0.04 | 0.029 |
| *Unclassified MollicutesRF9* (*Order*) | 0.32[b] | 0.43[b] | 0.41[b] | 2.00[a] | 0.185 | <0.001 |
| *Succinivibrionaceae UCG-002* | 0.50[b] | 0.70[a] | 0.66[a] | 0.72[a] | 0.029 | 0.02 |
| *Ruminiclostridium 9* | 0.45[ab] | 0.74[a] | 0.27[b] | 0.19[b] | 0.064 | 0.004 |
| *Ruminobacter* | 0.25[b] | 0.39[a] | 0.40[a] | 0.40[a] | 0.022 | 0.032 |
| *Subdoligranulum* | 0.29[a] | 0.28[a] | 0.28[a] | 0.18[b] | 0.017 | 0.044 |
| *Unclassified Gastranaerophilales* (*Order*) | 0.03[b] | 0.14[a] | 0.08[ab] | 0.14[a] | 0.017 | 0.039 |
| *Ruminiclostridium 1* | 0.15[a] | 0.08[ab] | 0.03[b] | 0.04[b] | 0.017 | 0.045 |
| *Ruminococcaceae UCG-001* | 0.04[b] | 0.09[a] | 0.10[a] | 0.08[a] | 0.007 | 0.014 |
| *Turicibacter* | 0.04[b] | 0.08[a] | 0.07[a] | 0.067[ab] | 0.005 | 0.024 |
| *Akkermansia* | 0.16[a] | 0.03[b] | 0.03[b] | 0.01[b] | 0.021 | 0.029 |
| *Desulfovibrio* | 0.04[b] | 0.10[a] | 0.03[b] | 0.01[b] | 0.010 | 0.011 |
| *Cetobacterium* | 0.03[b] | 0.02[b] | 0.04[ab] | 0.07[a] | 0.006 | 0.022 |
| *Lachnospiraceae NK4B4 group* | 0.10[a] | 0.02[b] | 0.01[b] | 0.02[b] | 0.012 | 0.020 |
| *Comamonadaceae other (Family)* | 0.01[b] | 0.03[a] | 0.02[ab] | 0.03[a] | 0.003 | 0.033 |
| *Novosphingobium* | 0.01[b] | 0.03[a] | 0.02[ab] | 0.01[b] | 0.003 | 0.011 |
| *Pseudomonas* | 0.01[b] | 0.005[b] | 0.007[b] | 0.03[a] | 0.003 | 0.042 |
| *Unclassified Oxalobacteraceae (Family)* | 0.01[b] | 0.03[a] | 0.01[b] | 0.01[b] | 0.003 | 0.007 |
| *Aerococcus* | 0.02[a] | 0.004[b] | 0.01[b] | 0[b] | 0.002 | 0.015 |
| *Massilia* | 0.004[a] | 0[b] | 0.004[a] | 0.02[a] | 0.002 | 0.030 |
| *Mycobacterium* | 0.004[b] | 0[b] | 0.004[b] | 0.01[a] | 0.002 | 0.003 |
| *Corynebacterium 1* | 0.002[b] | 0.004[ab] | 0.01[a] | 0.002[b] | 0.001 | 0.049 |
| *Bryobacter* | 0[b] | 0[b] | 0[b] | 0.01[a] | 0.001 | 0.022 |
| *Lachnoclostridium 1* | 0[b] | 0.01[a] | 0[b] | 0[b] | 0.001 | 0.022 |

**Table 2** (*continued*)

| Item[1] | Treatments[2] | | | | SEM | *P*-value[3] |
|---------|-----------|-----------|----------|---------|-----|---------|
| | **2,500 μm** | **1,000 μm** | **100 μm** | **10 μm** | | |
| *Gelria* | 0[b] | 0[b] | 0[b] | 0.01[a] | 0.001 | 0.010 |
| *SHA-109 other (Phylum)* | 0[b] | 0[b] | 0[b] | 0.01[a] | 0.001 | 0.010 |
| **Archaeal genus** | | | | | | |
| *Methanobrevibacter* | 62.48[c] | 75.93[b] | 89.68[a] | 90.40[a] | 2.419 | <0.001 |
| *Methanosphaera* | 35.47[a] | 23.04[b] | 8.39[c] | 8.26[c] | 2.392 | <0.001 |

**Notes.**

The "0" represent not detected;

[1] Data are the means of six replicates;

[2] The particle size of alfalfa was 2,500, 1,000, 100 and 10 μm; data are the average of relative abundance;

[3] Values with different superscripts in the same row mean significant difference ($P < 0.05$).

results are supported by a number of existing studies (*Bäuerl et al., 2014*; *Combes et al., 2011*; *Zhu, Wang & Li, 2015*). *Monteils et al. (2008)* reported that the Firmicutes in the rabbit caecum contain a large number of fiber-decomposing bacteria, this coincides with the fact that rabbits are adapted to high-fiber diets. Same as previous research (*Zhu, Wang & Li, 2015*), the abundance of Bacteroidetes (23.96%~36.05%) is the second largest bacterial community in our study, it is indicated that Bacteroidetes do play an important role in intestinal digestion in rabbits. Research has suggested that there were high abundance of Bacteroidetes in the intestine of herbivore animals related to the higher fiber content of food intake (*Crowley et al., 2017*). Contrarily, we found that the relative abundance of Firmicutes and Bacteroidetes were slightly different from previous studies (*Arrazuria et al., 2016*; *Jin et al., 2018*). One of the reasons maybe the results of sex selection in rabbits (*Arrazuria et al., 2016*). Moreover, it may also be the fact that the individual cages was not used in this work as previous reports (*Jin et al., 2018*). Certainly, it was possibly caused by feeding coccidiostatic as well. But the specific reasons for the differences still need to be proved by further experiments.

Proteobacteria are commonly found in the gastrointestinal tract of animals and exist as the dominant bacteria of some animals (*Jami et al., 2013*; *Fang et al., 2012*). In the present study, Proteobacteria (2.69% ~3.37%) was the dominating bacteria in rabbit caecum after Firmicutes and Bacteroidetes. At the same time, we observed that the relative abundance of Proteobacteria was significantly increased with the decrease of alfalfa particle size. This may be due to the relative increase of crude protein content or more protein binding site in the diet when the particle size of alfalfa decreased (Table S1). This was in line with the opinion that Proteobacteria was related to protein digestion (*Jami et al., 2013*; *Liu et al., 2016*). As the main genus of Proteobacteria, the relative abundance of *Succinivibrionaceae UCG-002* was significant increased with the reduction of particle size of alfalfa. Therefore, we assumed that *Succinivibrionaceae UCG-002* might be a microorganism involved in protein digestion. But further work is needed to confirm this speculation. In addition, it was more interesting that the relative abundance of Tenericutes (2.11%) in the 10 μm group was significantly higher than the other three groups and became one of the major microbe of the caecum. Nevertheless, *Arrazuria et al. (2016)* found that the relative abundance of Tenericutes only was 0.43% in the caecum of regular diet fed rabbits, which was similar to the Tenericutes
abundance of the other three treatments in this study. Therefore, it is regarded that the increased relative abundance of Tenericutes was attributed by the ultrafine smashing of alfalfa. It is a pity that the specific role of Tenericutes in the cecum of rabbits remains unrevealed.

In terms of genera, Unclassified *Bacteroidales S24-7* and *Ruminococcaceae UCG-014* were the two most abundant bacteria in rabbit caecum. However, *Bäuerl et al. (2014)* reported that the most frequent genera in healthy rabbits were *Ruminococcus* and *Alistipes*. This might have resulted from the different rabbit species and diets used in two studies. It was reported that the specific role of Unclassified *Bacteroidales S24-7* in the intestine was to breakdown starch, complex polysaccharides and fiber (*Lan et al., 2006*). Studies have shown that excessive oxalate can cause kidney stones by complexing calcium (*Whiteside et al., 2015*), while Unclassified *Bacteroidales S24-7* can degrade oxalate (*Ormerod et al., 2016*). This further confirmed the important role of Bacteroidetes in the cecum of rabbits. Furthermore, previous studies have shown that family Ruminococcaceae was closely related to fiber degradation (*Wood, 1988*; *Ezaki, 2015*). And that the *Ruminococcaceae UCG-014* as the main genus of *Ruminococcaceae*, should also play a role in fiber degradation. This is consistent with the common sense of rabbits as a herbivore.

*Lactobacillus* as a beneficial microorganism can inhibit the growth of pathogenic bacteria, improve the structure of intestinal flora, strengthen the intestinal barrier function, thereby enhancing the immunity of host (*Wang et al., 2015*). It was well documented that feed particle size has various effects on the growth of beneficial bacteria such as intestinal *Lactobacillus*. *Bao et al. (2016)* found that the number of *Lactobacillus* in the intestine of pigs increased first and then decreased as the size of corn crushed increased. *Singh et al. (2014)* evidenced that caecal *Lactobacillus* increased linearly with the increase of corn grinding size. Contrarily, our study found that the abundance of *Lactobacillus* increased with the decrease in particle size of alfalfa hay. This may be due to the slowdown of chyme passage when the particle size of fibers decreased, thereby enhancing the adhesion of Lactobacillus (*Pickard & Stevens, 1972*). Our previous study (*Liu et al., 2018*) also found that the ADG and ADFI of rabbits increased significantly, while the FCR decreased with the decrease of alfalfa particle size. This indicates that *Lactobacillus* maybe have a positive effect on the growth performance of rabbits, this was also been confirmed by *Wang et al. (2017)*. In addition, only *Ruminococcaceae UCG-014* and *Lactobacillus* (Firmicutes) had the same changes trend among the core genera. Furthermore, with the exception that *Ruminococcaceae UCG-005* was increased first and then decreased as the particle size of alfalfa decreased, the abundance of other genera (abundance >1%) of Firmicutes (mainly *Lachnospiraceae NK4A136 group* and *Ruminococcaceae NK4A214 group*) were significantly lowered when particle size decreased. This led to no statistical difference in the abundance of Firmicutes among treatments.

Methanogens belong exclusively to the domain of archaea. They play a key role in the final stage of microbial organic matter decay in the digestive system (*Cavicchioli, 2011*). Thus, the methanogenic archaea are widely found in the gastrointestinal tract of herbivores and many methanogens have been isolated and identified from different animals (*Jin et al., 2017*; *Wright et al., 2004*). However, almost all of the archaea found in the gastrointestinal

tract of animals originated from Euryarchaeota (*Jin et al., 2017*; *Zhu et al., 2016*). This was consistent with our study. The results of our work showed that the Euryarchaeota almost covered the whole archaeal phyla in all treatment groups (over 99.9%). Therefore, we believe that the Euryarchaeota plays a major role in the rabbit caecum.

It is reported that the archaeal community in rabbit caecum is unique and of low complexity with few dominating species, *Methanobrevibacter* and *Methanosphaera* genus usually were the two most abundantly distributed genera (*Zhu et al., 2016*; *Kušar & Avguštin, 2010*). In particular, the genus *Methanobrevibacter* was considered to be the absolute dominanting core genus (*Wright et al., 2004*; *Zhu et al., 2016*), which is in agreement with the findings of current study. Researches proved that *Methanobrevibacter* and *Methanosphaera* were two dominanting $H_2$-consuming organisms that were often found distributed in the hindgut of humans or animals, catalyzing the conversion of hydrogen and carbon dioxide, methanol, etc. into methane (*Kušar & Avguštin, 2010*). Here, we discovered that fed alfalfa with different particle sizes could adjust the abundance and diversity of methanogens. As the alfalfa particle size decreased, the relative abundance of *Methanobrevibacter* increased at the cost of a reduction in *Methanosphaera*. This was confirmed by our team, that when the relative abundance of *Methanobrevibacter* increased the methane production increased (*Liu et al., 2018*). This phenomenon could be explained by the two aspects. At one hand, some earlier studies found that the particle size of fibrous ingredients was known to affect retention time of digesta in the intestine (*Gidenne et al., 1991*; *Gidenne, 1993*). The smaller the fiber particle size, the longer retention time of the feed in the intestine. This prolongs the fermentation time of the feed in the caecum, resulting in a larger amount methane production. On the other hand, compared to *Methanobrevibacter*, the production of methane by *Methanosphaera* has certain limitations. *Methanosphaera* can not produce methane from hydrogen and carbon dioxide, formate, acetate, but can only use methanol and hydrogen to produce methane, which requires the participation of ATP (*Fricke et al., 2006*; *Miller & Wolin, 1985*). *Methanosphaera* and *Methanobrevibacter* consume 1.3 moles of methanol and 1 mole of carbon dioxide, respectively, to produce 1 mole of methane (*Fricke et al., 2006*; *Hook, Wright & McBride, 2010*). This means that the capacity of *Methanobrevibacter* to produce methane is larger than that of *Methanosphaera*. Consequently, exchange abundance of *Methanosphaera* and *Methanobrevibacter* increased methane production.

### Nucleotide sequence accession numbers

Sequences of this project have been deposited into the Sequence Read Archive (SRA) of the NCBI nucleotide database under accession number PRJNA542420.

## CONCLUSION

In summary, as the particle size of alfalfa meal decreased, both the bacterial and archaeal population in the caecum of rabbit experienced variations. However, the changes in the archaea are produced earlier in time. The bacterial populations changed when the alfalfa fiber particle size decrease from 100 to 10 μm, whereas the archaeal populations changed while the fiber particle size decrease from 1,000 to 100 μm. In terms of bacteria,

*Ruminococcaceae UCG-014* and *Lactotobacillus* increased whereas the *Lachnospiraceae* NK4A136 group, *Ruminococcaceae* NK4A214 group decreased when the fiber particle size decreased from 2,500 to 10 μm. As for the archaeal populations, the *Methanobrevibacter* increased at the cost of redution of *Methanosphaera.* The gastrointestinal microbial populations could be manipulated by feeds processing technology in the aim of promoting animal production performance.

### Funding
This work was supported by the National Natural Science Foundation of China (31402104). The funders had no role in study design, data collection and analysis, decision to publish, or preparation of the manuscript.

### Grant Disclosures
The following grant information was disclosed by the authors:
National Natural Science Foundation of China: 31402104.

### Competing Interests
The authors declare there are no competing interests.

### Author Contributions
- Mei Yuan performed the experiments, analyzed the data, prepared figures and/or tables, authored or reviewed drafts of the paper, approved the final draft.
- Siqiang Liu performed the experiments, prepared figures and/or tables, approved the final draft.
- Zhisheng Wang, Bai Xue, Gang Tian and Jingyi Cai conceived and designed the experiments, authored or reviewed drafts of the paper, approved the final draft.
- Lizhi Wang and Huawei Zou conceived and designed the experiments, prepared figures and/or tables, approved the final draft.
- Quanhui Peng conceived and designed the experiments, contributed reagents/materials/analysis tools, authored or reviewed drafts of the paper, approved the final draft.

### Animal Ethics
The following information was supplied relating to ethical approvals (i.e., approving body and any reference numbers):

The Ethical Committee of the Sichuan Agricultural University provided full approval for this research (SYXK(chuan)2014-187).

### Data Availability
Sequences of this project are available at the Sequence Read Archive (SRA) of the NCBI nucleotide database: PRJNA542420.

## Supplemental Information

Supplemental information for this article can be found online at http://dx.doi.org/10.7717/peerj.7910#supplemental-information.

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
