# Peer review of "Effects of particle size of ground alfalfa hay on caecal bacteria and archaea populations of rabbits"

_PeerJ, doi:10.7717/peerj.7910_

## Round 0.1 · original submission · Major Revisions

Your manuscript has now been assessed by three reviewers. While they found merit in the work, there are a number of issues that need clarification.

In particular, reviewer 3 has noted that the experimental design and data in this paper appear very similar to a paper by the same author published in Plos One last year. Having inspected the Plos One paper myself, I find it to be remarkably similar, including the results of increases in the proportion of Methanobrevibacter as particle size of the feed decreased. I request that you state clearly in your response how this submission differs from the previous paper, including whether the sequencing data and experimental animals used were the same.

From my own reading of the paper, I request that you include more detail on your statistical analysis, including clearing up the R/SAS issue. In particular, I ask you include the actual model specification used for your mixed models, i.e. what variables were the response, predictors and random effects? This will ensure transparency and repeatability.

Xavier Harrison


Reviewer 1 ·

Basic reporting

Reporting language is clear although English editing is encouraged. The type of corrections that should be made are minor but mandatory. Some examples of corrections I suggest are:

Line 9. “treatment groups” instead of “treatments group”.
Lines 27. “experienced” instead of “experience” and “archaea are” instead of “the archaea is”.
Line 33. “herbivore animal that has” instead of “herbivore animals that have”.
Line 41. “and so on” does not sound very scientific.
Line 63. “In addition, most” instead of “In addition, the most”
Line 67. “in rabbits exist” instead of “in rabbits exists”

More corrections of this type are needed in the rest of the manuscript so I reccomend careful English editing.

Line 75. Animal experimentation permission code should be included.

Table S1. I find the nutritional data redundant because percentages are identical in all four formulations but it is OK to leave it this way since it is supplementary.

Literature references are sufficient. The article is well structured and raw data es adequately shared.

Experimental design

I find the study of interest. The effect of particle size should be explored.
The experimental design is original and perhaps this data combined with previous results of other parameters within the same experiment would be more complete.
Methods. Animal and experiment sample collection. How was the intake of alfafa controlled? Was first feeding always consumed before second feeding was admisnistered? If not, how was second feeding administered? Were animals individually caged? How was slaughter performed? Anesthetics drugs and doses should be detailed.
DNA extraction. Raw data on DNA quality and concentration should be included as supplementary material (a table with the 120 values of concentration and 260/280 ratio) or at least in the results section DNA quality should be mentioned.
PCR amplification, Illumina library
Was control library spiked in amplicon pools in order to improve the unbalanced and biased base composition of the generated 16S rRNA libraries? If so, please report it.

Validity of the findings

Findings are interesting. I would like conclusions to be better fundamented. I don't understand why authors state that archaea are more sensitive to changes thatn bacteria. This is not the picture I get whean I look at Figures 5 and 6 I see more phylum variability among treatment groups in bacteria than in archaea. Am I missing something? Could it be better explained?

I would work on the figures as indicated below:

Figure 4. Results for effect of small particle on bacteria and large particles in archaea seem homogenous. The rest are disparate. This is not really discussed. The authors state that their PCoA analysis reveals that archaea are more sensitive than bacteria. From the PCoA I interpret that intersample variability is greater among archaea when fed with small size particles.

Figure 5. The legend is confusing because there are far more groups in the legend than in the bars. Please adapt.

Figure 6.The same as Figure 5.

Additional comments

The study reported by Yuan and collaborators on the effect of particl size of ground alfafa hay on caecum bacteria and archaeas populations of rabbits shows interesting results. However, there is some information missing in methodology and conclusions should be better supported. Please address my concerns.

·

Basic reporting

This manuscript presents a novel data in relation to rabbit caecal microbiota (bacteria and archaea) and alfalfa particle size. The article is well structured with enough figures, tables and the raw data are shared. Although some parts are well written using clear and professional English, there are other parts in the manuscript with substantial English mistakes. Although I collect some of them, I would like to suggest to the authors to revise it thoroughly.

L42- “function in animal”, instead “animals”
L62-“specific studied on changes in microbial composition and structure is scarce” “specific studies on changes in microbial composition and structure are scarce”
L63-“the most” instead “most”
L81- “the animal was fed twice daily at 0900 and 01700” instead “the animals were fed twice daily at 09:00 and 17:00”.
L94- “the extracted DNA were detected” instead “was detected”
L99- “total DAN” instead total DNA”
L170- “In total of 2107 OTUs were shared in all of four treatments” instead “A total of 2107..”
L174- “In total of 2246 OTUs were assigned” instead “A total of 2246…”
L186-“bacterial composition is changed when..” instead “bacterial composition changed”
L191-“that the archael composition” instead “archaeal”
L215-“with relative abundance were larger than 1%” instead “with relative abundance larger than..”
L220-“with relative abundance was above 1%” instead “with relative abundance above..”
L244- “Table 2 showed” instead “Table 2 shows”.
L270: “Furthermore, bacteria belongs to the” instead “bacteria belonging to the..”
L279: “animals are related to the higher fiber content of food intake” instead “animals related to the..”
L334: “it was reportd that” instead “it was reported that”.


The cited references are accurate and a sufficient field background is provided. However, in the L31-33 the author state that rabbit meat is important in china. I would like to suggest to include some specific data about rabbit meat production or consumption to put in context the importance of this sector in some countries.


The presented Figures are relevant to the content of the article and appropriately described and labeled. However, since in the rarefaction curve (Figure 1) are 24 curves (samples) included, I would like to suggest representing the OTU mean of each group (10,100,1000,2500) with the SD or SEM of each group. In the Figure 2 there is typo in “OUTs”. It would be helpful also, have a better-quality Figure 6 to be able to read properly the taxonomic groups. In addition, in the Table 1, the used superscripts are confusing. In the Shannon value are some numbers with b superscript at different size and later is stated that the particle size of alfalfa meal was 2500, 1000, 100 and 10 μm, respectively. It is confusing. If the objective of the author is to indicate the significant differences I would suggest mark the p value with an asterix or use any number as superscript with further explanation. As in the Table 1, in the Table 2 the superscripts are very confusing as well. Superscript a and b are refereed to an item and treatments, but they are also used in the given numbers.

Experimental design

The used methods in this study are the appropriate ones to try to answer the stated hypothesis. The animal experiment has been approved for an animal care committee. However, it would be convenient if the authors refer the specific number or reference of the approved protocol (L72).

In the Line 81 the author say that animals were fed twice daily and later (L83-84) is stated that animals had ad libitum access to the diet. It is confusing, and some clarification of what food was limited or ad libitum is required.
In addition, there is some missing information about experimental design like where the animals were housed (research facility or experimental farm), the sex of the animals, and how they were housed, by groups or individually (specially the animals that were sampled). The information of how animals were slaughtered and if all animals were slaughtered the same day of on groups in consecutive days is missing. Also, the sampling procedure is not thoroughly explained. Since this is a microbiota study it should be highlighted how the cross contamination was avoided and how a representative sample from the whole caecal content was obtained.

Regarding the sequencing I wonder if in the sequencing run any negative or positive control was included. The positive and negative controls help detecting cross contaminations during the library preparation or a quality control for the sequencing process.


Following the recommendations for 16SrRNA sequencing I assume that the authors did include PhiX control in the run. If so, it should be reflected in the manuscript along with the used concentration. In the materials and methods section, singeltons are defined as a “bad amplicons” (in the bioinformatic analysis section), while I would define them more as a sequencing errors as the reference provided. Qiime version used should be stated as well as Silva database used to assign taxonomy. I addition, I wonder why the author used Mothur software to calculate diversity indexes if it can be done in Qiime. Is there any scientific reason?

In the statistical analysis the author state that they used MIXED procedure in R. It may be a mistake considering that in general mixed procedure is a SAS procedure rather than R (although it could be implemented within some R packages). Anyway, it should be indicated the specific R packaged used for data analysis. Also, the specific statistical analysis implemented for each data comparison (in R or SAS software) should be provided.

The author only presented the analysis of the weighted UniFrac distances that refers to a quantitative analysis. Were the unweighted UniFrac distances analyzed (qualitative analysis)? Is there any reason to do not include the analysis of unweighted UniFrac distances? If they were analyzed, and no significant differences were detected it should be mention in the manuscript.
Regarding the PCoA representation of the UniFrac distances (Figure 4), I strongly encourage the authors to complement the analysis with a statistical analysis to test whether the observed differences in the plot (clusters) are statistically significant (for example using Permanova test).

Validity of the findings

This study collected a meaningful result. Although some details are lacking in the methodology the obtained data look robust. However, I would like to suggest to the authors avoid using the word “significant differences” (L187) in the results if a p value is not provided.
In addition, in the L-207 is stated that “the archaeal composition structure changed before the bacterial structure”. Since all animals are sampled at the same time the data are not providing enough information to make this statement and it should be removed.
In the manuscript there are some references to anaerobe instead archaea (L157, L194). Anaerobe refers to oxygen requirement and booth archaea and bacteria can be aerobic or anaerobic. Therefore I suggest to change anaerobe by archaea.


Conclusion are well stated and linked to the original research question. The authors attempt to describe the impact of fiber size on the caecal microbiota. However, this manuscript is a complementary study (same experiment) to a previously published study, assessing production performance, digestibility and methane emissions. (Liu, S.et al. 2018. PloS one). Therefore, the authors should take advantage of the available data (although previously published) and link the microbiota results with the production performance (there is only a small reference to methane production in the discussion).

In the discussion there is no any reference to the effect or particle size in microbial diversity. Although generally high diversity has been claimed as beneficial, in the case of ruminants, lower richness of microbiome gene content and taxa was tightly linked to higher feed efficiency (Shabat, et al.The ISME journal 10.12 (2016): 2958). In the present study there are significant differences in the Shanon index between treatments and the authors have published the productive data. Therefore add this point to the discussion would enrich the manuscript content.

In the L-282 states that Proteobacteria “exist as the dominant bacteria of some animals”, please provide a reference.

In the discussion is stated that (L294) “it is speculated that the increased relative abundance of Tenericutes may be attributed by the ultrafine smashing of alfalfa”. I am lacking the information of its impact in rabbit’s performance (it is good or not?) and if there is more literature in this regard.
Lactobacillus is generally present in low abundance in the cecum of rabbits, and the factors affecting the colonization has been studied (ie. Yu, B., et al. 1993. Journal of applied bacteriology, 75(3), 269-275). In the present study data suggest that small fiber particle size could increase its presence. How could the particle size enhance lactobacillus colonization? Is the presence of Lactobacillus associated with better growth parameters?


I found difficult to understand the following sentences (L209-211): “ Species annotation and statistics were performed on different taxonomic levels using the effective sequences obtained (Fig. 5). The results showed that the sequences were finally annotated as bacteria (99.87%) and archaea (0.13%)”.
I would like to suggest to the authors to express it in a different way taking into account that the sequencing process is carried out after 16S rRNA amplification (PCR) and is difficult to extrapolate the amount of sequences obtained in bacteria and archaea if they have been amplified (before sequencing) with different primers.
In addition I found difficult to understand L226-230. The author should rephrase it.

Regarding the conclusion section, most of its content is a summary of the obtained results and it should be reflected at the beginning of the paragraph (Summarizing,…..) to do not confuse the readers.

In addition, I would like to suggest to the authors to include in the last part of the conclusion what would be the benefit of manipulate the gastrointestinal microbial population by feeds processing technology (Improve animal health, growing performance, increase farms profit, decrease meat price, etc...).

Reviewer 3 ·

Basic reporting

The manuscript submitted by Yuan and coworkers describes the characterization of rabbit cecal microbial communities in response to different particle sizes in their diets using Illumina amplicon sequencing. The authors show that a reduction in particle size leads to changes in microbial community composition.

My main concern is that is unclear how much this study overlaps with another recent study from the same lab: Liu, et al, 2018 PlosOne (as stated in their submission). The authors need to clearly highlight and state potential overlaps and differences between the 2 studies which obviusly seem to have used the same animal trial (and sequencing samples?). Otherwise, I fear double-publication of the same data, without clarification of this issue. The authors should either refer to and discuss their current resutls with those from the 2018 PlosOne paper, or mention why there is no overlap between the 2 studies. Specifcally, are these e.g. the same sequencing data? On the other hand, if these two studies are sufficiently different from eah other, the authors should discuss the results e.g. from methane production in the currently submitted manuscript.

The authors should improve English usage throughout the manuscript. There are a number of minor English errors in grammar (e.g. singular/plural - e.g.: animal-animals, rabbit-rabbits, microoganism-microorganisms, genus-genera). I suggest that the authors have a more experienced English speaker help with English usage.

I would like to encourage the authors to provide more details about the rabbit microbiota in general and rabbit microbiome studies in the introduction section.

It seems that the raw sequencing data have not been submitted to NCBI SRA, in case the data have not been submitted, please submit the sequencing data to NCBI SRA.

Experimental design

no comment

Validity of the findings

no comment

Additional comments

Specific comments:
L3: should be "communities"
L33-34: Plese reword and use either singular or plural consistently.
L34-41: Please mention that rabbits are hindgut fermenters.
L57: Please change "Flexibacter-Cytophaga-Bacteroidetes" to: "Bacteroidetes"
L58: Should be "albus"
L99: Should be: "16S rRNA gene", and "DNA"
L113: please change "strip" to "band"
L124-146: Did the authors perform a chimera check?
L133: should be "lead"
L157: Please clarify: do the authors refer to archaea with "anaerobe"?
L158 (and Table 1): Were the samples normalized to the same number of reads for alpha diversity estimations?
L180: Please rephrase this sentence, it is unclear.
L185: Please rephrase "crossed"
L228: Please change "left" to: "remaining 10 were". Should be: "The number of genera with relative..."
L260: Please rephrase "attempted"
L270: Should be "belonging"
L277: please rephrase "community" - Bacteroidetes was e.g. the second most abundant phylum or group. L278-280: Indeed Bacteroidetes is among the most abundant bacterial phyla in animal microbiota (please rephrase "suggested". Please provide some references here. I suggest that the authors expand their discussion here and discuss a potential role of Bacteroidales S24-7/Mutibaculaeae phylotypes here. See e.g. Lagkouvardos et al., 2019 "Sequence and cultivation study of Muribaculaceae reveals novel species, host preference, and functional potential of this yet undescribed family" or Ormerod et al., 2016: "Genomic characterization of the uncultured Bacteroidales family S24-7 inhabiting the guts of homeothermic animals" for more details (please see also L300-302).
L281-282: Please delete "as gram-negative bacteria"
L282: Please reword "dominating" here
L284-286: Were any proteobacterial genera significantly increased in abundance in response to decrease particle size? If so, please briefly discuss.
L288: Please rephrase "declaim"
L307 (also L316): Please rephrase "probiotic" - in this case, Lactobacillus was not added as a probiotic to the diet, I assume. Maybe "potential beneficial"?
L352-356: Please clarify: what do you mean by "limitations" - a reduced/lower methane production? Please disucss the increase of Methanosphaera in more detail. Do the authors claim that an increase of Methanospaera (and a decrease in Methanobrevibacter) would be beneficial because of lower methane production levels?

Fig. 1: I suggest to move Fig. 1 to the supplementary material
Fig. 2: (Figure legend) please correct "OUTs"
Fig. 5: I suggest to either delete panel B or move it to the supplementary material. Because of the dominance of just one phylum, I think this panel is not very informative. Should be "communities"
Fig. 6: Should be "ceacal" and "communities"

Table 2: The title should read: Phyla and genera with different abundance in ceaca of rabbits fed diets with different particle sizes of.... Please clarify: What is shown in the "treatments" columns - mean/median relative abundances?

---

## Round 0.2 · Minor Revisions

Many thanks for making the required revisions to your manuscript. Your paper has now been reassessed by the original reviewers, and there are still some minor issues that need clarification.

In particular, please could you provide more detail on the methodological approaches for your library preparation as highlighted by reviewer 2. In addition, reviewer 2 has provided some suggestions for alterations to the statistical reporting that I would like to see carried out.

Both reviewers have highlighted some inconsistencies in the phrasing of written text. Please update the manuscript in line with reviewer 1's helpful suggestions for improved clarity. I'm happy to have a look over the revised version to clear up any other inconsistencies that remain.

Reviewer 1 ·

Basic reporting

Reporting language is clear although English editing in the new information that has been added, is encouraged. Authors have correctly addressed concerns. However, minor corrections should be incorporated prior to definetly accepting the paper. These are mainly in the new modified parts of the manuscript and are the following:

Lines 102-103 Rephrase “Once slaughtered, bring sterile gloves to collect cecal 50 g contents to sterile test tubes.” to “Once slaughtered, 50 g of cecal content was collected in sterile conditions.”

Line 112. Eliminate simple form the word buffer. “was eluted from the elution column with TE buffer”

Lines 317-18. Verb not in correct tense and author’s name in reference is incorrect. Please change “There are also studies showed that high diversity of microorganisms was beneficial to animals (Aeeazaria et al., 2016).” to “There are also studies showing that high diversity of microorganisms is beneficial to animals (Arrazuria et al., 2016).

Lines 364-365 Change “It is further confirmed the important role of Bacteroidetes in the cecum of rabbits.” to “This further confirms the important role of Bacteroidetes in the cecum of rabbits.”

Lines 440-441 I suggest changing “However, the archaea is changed earlier than bacteria as alfalfa meal particle size decreased. To “However, these changes in the archaea are produced earlier in time”

Experimental design

No comment

Validity of the findings

No comment

Additional comments

No comment

·

Basic reporting

The manuscript has been improved substantially. However, still some English editions are needed because there are some sentences like “archaea responsed earlier”, “There are also studies showed”, “However, the archaea is changed earlier than bacteria as” etc..

In addition, I would like to suggest to the author separate the sentence in L 66-71 in two sentences, because that long sentence it is difficult to understand. Also, the ADG and ADFI acronyms should be defined in the text.

Experimental design

The author included information about the sex of the animals included in the whole experiment. But it is not clear if it was a sex-selection made for sampling. If the same number of samples from male and females were used for sequencing was there any difference in the microbiota based on the gender that could impact the obtained results?
Have the authors showed any “cage effect” in the microbiota? It has been described in mice and it would be interesting to know at what extent this factor affects rabbits microbiota.
In any case, sex and cage effects should be mention in the discussion, especially when the results are compared with previous studies.

The authors introduce in L 133 that they used emulsion PCR. What was the goal of that PCR? Could you provide more details about it? Afterward, was the electrophoretic band of the emulsion PCR purified?

In addition in the L138, is stated that TruSeq DNA PCR-Free Sample Prep Kit following manufacturer’s recommendations was used for library preparation. However Illumina recommend this kit for whole genome sequencing and no for amplicon sequencing. What was the reason for using this library preparation kit? It has been previously validated for amplicon sequencing?

https://www.illumina.com/products/by-type/sequencing-kits/library-prep-kits/truseq-dna-pcr-free.html

As in the first review, I would like to suggest again change the term “bad amplicon” referring to the singletons for “sequencing errors” or at least if the author does not consider it is appropriated it should be explained why is considered “bad amplicon”.


Although the formula of the MIXED procedure of SAS has been included, I Would like to suggest to the author to be more specific about the R packages, comparison test and multiple comparisons corrections used for ALL data analysis.

I would like to suggest to the author to include the statistical differences sowed by permanova analysis in the PCoA figure and/or include in the supplementary table 2 what are the analyzed data.
For a complete analysis of the UniFrac distances the weighted as well the unweighted distances should be analyzed in order to understand if the differences are only quantitative or both qualitative and quantitative. I would like to suggest again to include both measurements in order to complement the UniFract distances analysis.

L236-241 I would suggest rephrasing this sentences taking into account that the PCoA does not demonstrate significant differences, they are obtained by Permanova analysis taken all samples. It seems confusing in the text (specially the exceptions).

Validity of the findings

As I stated before the cage effect as well the sex effect could have an impact on the obtained results and they should be mention in the discussion and taken into account when comparing the obtained results with previous studies.
In addition the used coccidiostatic could have an impact on the observed differences with a previous study (stated in L352) and it should be mention.

---

## Round 0.3 · Minor Revisions

I have now assessed the revised manuscript and am largely satisfied with the revisions. I would just ask for more clarity on the issue of the emulsion PCR for library preparations.

The methods read as if your library prep was a 2 step process with 'conventional PCR' to generate 200-400bp bands and then emPCR for further amplification. Is this correct - is the emPCR the barcoding step? If so I think a summary sentence in the manuscript would help greatly here for context. Something like:

"We used a 2 step process for library preparation. First, 16S amplicons were generated using PCR to generate 200-400bp amplicons. [text on PCR conditions]. Second, we added unique barcodes to samples using emulsion PCR. [text on benefits of emPCR and reducing chimera formation]."

In addition, could you provide a reference for the reduction of chimeras by emPCR? (line 153)

---

## Round 0.4 · accepted · Accept

Thank you for making the required changes to the manuscript. I am now happy to recommend it be accepted for publication.